# The validity and reliability of observational assessment tools available to measure fundamental movement skills in school-age children: A systematic review

Lucy H. Eddy[1,2,3]*, Daniel D. Bingham[2,3], Kirsty L. Crossley[2,3◉], Nishaat F. Shahid[2◉], Marsha Ellingham-Khan[2,3‡], Ava Otteslev[2,3‡], Natalie S. Figueredo[2,3], Mark Mon-Williams[1,2,3,4], Liam J. B. Hill[1,2,3]

1 School of Psychology, University of Leeds, Leeds, West Yorkshire, United Kingdom, 2 Bradford Institute for Health Research, Bradford Royal Infirmary, Bradford, West Yorkshire, United Kingdom, 3 Centre for Applied Education Research, Wolfson Centre for Applied Health Research, Bradford Royal Infirmary, Bradford, West Yorkshire, United Kingdom, 4 National Centre for Optics, Vision and Eye Care, University of South-Eastern Norway, Notodden, Norway

◉ These authors contributed equally to this work.
‡ These authors also contributed equally to this work.
* L.Eddy@leeds.ac.uk

## Abstract

### Background

Fundamental Movement Skills (FMS) play a critical role in ontogenesis. Many children have insufficient FMS, highlighting the need for universal screening in schools. There are many observational FMS assessment tools, but their psychometric properties are not readily accessible. A systematic review was therefore undertaken to compile evidence of the validity and reliability of observational FMS assessments, to evaluate their suitability for screening.

### Methods

A pre-search of 'fundamental movement skills' OR 'fundamental motor skills' in seven online databases (PubMed, Ovid MEDLINE, Ovid Embase, EBSCO CINAHL, EBSCO SPORTDiscus, Ovid PsycINFO and Web of Science) identified 24 assessment tools for school-aged children that: (i) assess FMS; (ii) measure actual motor competence and (iii) evaluate performance on a standard battery of tasks. Studies were subsequently identified that: (a) used these tools; (b) quantified validity or reliability and (c) sampled school-aged children. Study quality was assessed using COnsensus-based Standards for the selection of health Measurement INstruments (COSMIN) checklists.

### Results

Ninety studies were included following the screening of 1863 articles. Twenty-one assessment tools had limited or no evidence to support their psychometric properties. The Test of

**Data Availability Statement:** All relevant data are within the paper and its Supporting Information files.

**Funding:** The work of the lead author (LHE) was supported by an ESRC White Rose Doctoral Training Partnership Pathway Award (ES/P000745/1). LJBH, MMW and DDB were supported by the National Institute for Health Research Yorkshire and Humber ARC (reference: NIHR20016), and the UK Prevention Research Partnership, an initiative funded by UK Research and Innovation Councils, the Department of Health and Social Care (England) and the UK devolved administrations, and leading health research charities. Weblink: https://mrc.ukri.org/research/initiatives/prevention-research/ukprp/. The views expressed in this publication are those of the author(s) and not necessarily those of the National Institute for Health Research or the Department of Health and Social Care. The work was conducted within infrastructure provided by ActEarly: a City Collaboratory approach to early promotion of good health and wellbeing funded by the Medical research Council (grant reference MR/S037527/). MMW was also supported by a Fellowship from the Alan Turing Institute. The funders had no role in study design, data collection and analysis, decision to publish, or preparation of the manuscript.

**Competing interests:** The authors have declared that no competing interests exist.

Gross Motor Development (TGMD, n = 34) and the Movement Assessment Battery for Children (MABC, n = 37) were the most researched tools. Studies consistently reported good evidence for validity, reliability for the TGMD, whilst only 64% of studies reported similarly promising results for the MABC. Twelve studies found good evidence for the reliability and validity of the Bruininks-Oseretsky Test of Motor Proficiency but poor study quality appeared to inflate results. Considering all assessment tools, those with promising psychometric properties often measured limited aspects of validity/reliability, and/or had limited feasibility for large scale deployment in a school-setting.

## Conclusion

There is insufficient evidence to justify the use of any observational FMS assessment tools for universal screening in schools, in their current form.

## Introduction

The importance of fundamental movement skills (FMS) has been well established with regard to children's development [1], but research reports a recent decline in the proficiency of children's FMS [2]. This is concerning as FMS are–by definition—foundational motor skills that underpin the development of more complex movement patterns required for participation in physical activity (bodily movement produced by skeletal muscles requiring energy expenditure) [3, 4]. The foundational nature of FMS means that they yield a broad-spectrum of associated benefits within childhood development [5]—including being positively associated with health, whereby children with well-developed FMS are more likely to participate in physical activity and have a lower body mass index [6–8]. Research has also found positive associations between FMS and education outcomes, including language and cognitive development, as well as attention and performance on standardised tests of academic attainment [6, 9–12].

The growing lack of proficiency in children's FMS is particularly disappointing as a recent systematic review of school-aged children found that FMS are consistently improved through training and interventions [13]. However, physiotherapists and occupational therapists are increasingly overwhelmed by the number of referrals for motor skill assessments [14], which has led to parental/guardian dissatisfaction with the services available to support children with motor skill difficulties [15–18]. The Chief Medical Officer has recommended the increased participation of schools in helping to reduce the burden on the National Health Service (NHS) in the UK [19]. The vision is for schools and healthcare services to collaborate and provide more community-based programmes and initiatives that enhance public health through increasing prevention and early identification of children in need of additional support. The need for such a collaboration has become yet more urgent after the Covid-19 crisis lockdown where many children have missed essential developmental experiences (e.g. playing outside and interacting with peers).

It can be seen that there are multiple potential benefits from the use of FMS assessments to screen all pupils within schools to identify those with poor FMS. It would encourage greater communication between families, schools and healthcare services, which has the potential to expedite access to treatment services and interventions [20]. It could help address health and educational inequalities attributed to socioeconomic status (SES) given that research from a large longitudinal cohort study found that mothers from a lower SES are less likely to access

primary care facilities [21]. It follows that children from a lower SES are less likely be identified as needing extra support with FMS development under current service provision, and therefore less likely to be offered intervention (at least within the UK). Universal FMS screening in primary schools would provide a more equitable approach to identifying those children in greatest need of support.

There are currently a large number of assessment tools used to measure FMS both clinically, and for research purposes. A large proportion of these assessment tools rely on an assessor observing children perform FMS on a battery of standardised tasks. Standardised observational measures are considered a useful way to assess children's FMS in schools [22] as they are reasonably low cost (relative to objective wearable sensors), have minimal data entry and analysis requirements for schools, and are also less susceptible to bias than proxy reports [23]. There are a large number of observational assessment methods being marketed to schools [22]. The saturation of such measures makes it difficult for teachers, practitioners, and researchers to know which assessment is best suited to identify accurately children who are struggling with FMS development. This evaluation is particularly challenging as there is a lack of clarity in the literature regarding the validity and reliability of the available observational measures.

A systematic review was required to document the psychometric properties of the observational assessment tools being promoted as measures of FMS to allow schools and health practitioners to make informed decisions about FMS assessment tools. This systematic review aims to: (i) establish a comprehensive summary of the observational tools currently used to measure FMS that have been subjected to scientific peer-review; (ii) examine and report the validity and reliability of such assessments.

## Methods

Methods for this systematic review were registered on PROSPERO (CRD42019121029).

### Inclusion criteria and preliminary systematic search

A preliminary search was conducted to identify assessment tools that were identified in peer-review published research as measures of FMS in school-aged children. This pre-search was conducted in the seven electronic databases (PubMed, Medline, Embase, CINAHL, SportDiscus, PsycInfo and Web of Science) in December 2018, and was subsequently updated in May 2020, using the search terms 'fundamental movement skills' OR 'fundamental motor skills'. Assessment tools identified in this pre-search were included in the subsequent review if they were confirmed to: (i) assess fundamental movement skills, including locomotor, object control and/or stability skills [24]; (ii) observationally measure actual FMS competence (i.e. physical, observable abilities); (iii) assess children on a standard battery of tasks which were completed in the presence of an assessor. Proxy reports and assessments that measured perceived motor competence were therefore excluded from the review. No restrictions were placed on the health/ development of included participants, as schools are faced with these issues, so any assessment tool that is going to be used in an educational setting would need to be appropriate for use with children both with and without developmental difficulties.

The titles and abstracts of the results of this pre-search were screened by the lead reviewer (LHE) to identify assessment tools mentioned within them that were being used to assess FMS. Any studies stating they were assessing FMS but omitting mention of the specific assessment tool in the title or abstract underwent a further full text review.

### Electronic search strategy and information sources

The search strategy developed (see S1 Table) was applied in seven electronic databases (PubMed, Medline, Embase, CINAHL, SportDiscus, PsycInfo and Web of Science) in January 2019, and was then updated in May 2020. Conference abstracts identified were followed up by searching for the full articles or contacting authors to clarify whether the work had been published.

### Study selection

For the initial search (Dec 2018), titles and abstracts were screened in their entirety by one reviewer (LHE), and two reviewers (NFS & KLC) independently assessed half of these studies each. The same process was followed for full text screening to identify eligible studies. Reviewers were not blind to author or journal information and disagreement between reviewers was resolved through consultation with a fourth reviewer (DDB). For the update, the same process was repeated with two different reviewers (ME-K & NSF, in place of NFS & KLC).

### Data extraction process & quality assessment

Three reviewers each extracted information from a third of the studies in the review in both the initial search (LHE, KLC & NFS) and the update (ME-K, AO & NSF). Data extraction and an assessment of the methodological quality of each study were completed using the Consensus-based Standards for the Selection of health Measurement INstruments (COSMIN) checklist [25], which outlines guidance for the reporting of the psychometric properties of health-related assessment tools. Information was extracted on: (i) author details and publication date; (ii) sample size and demographic information related to the sample; (iii) the assessment tool(s) used; (iv) the types of psychometric properties measured by each study; (v) the statistical analyses used to quantify validity or reliability and whether they were measured using classical test theory (CTT) or item-response theory (IRT); (vi) the statistical findings. Methodological quality ratings for each study were recorded as the percentage of the standards met for the included psychometric properties and generalisability. When an IRT method was used, a second quality percentage was calculated, based on the COSMIN guidelines for IRT models [25]. The lead reviewer (LHE) and a second reviewer (AO) each evaluated half of the studies for methodological quality, with a 10% cross-over to ensure agreement. Agreement was 100%, so no arbitration was necessary.

### Interpretation of validity and reliability

Many studies used different terminologies to describe the same type of validity or reliability, so it was necessary to set a definition for each psychometric property and categorise study outcomes in accordance to the COSMIN checklist [25] (see Table 1). Interpretability and face validity (sub-section of content validity) were not included as these could not be quantified using statistical techniques. Responsiveness was not included, as this is recognised as being separate to validity or reliability within the COSMIN guidance.

Due to a large variation in the statistical tests used to assess validity and reliability, a meta-analysis was not possible. To enable ease of interpretation of studies that utilised statistical analyses, a traffic light system was used (poor, moderate, good and excellent; see Table 2), which allowed such results to be grouped into different bands according to thresholds for these statistical values suggested in previous research. The results of all outcomes which utilised other statistical tests are described in the text. For the studies that included multiple metrics for each psychometric property, the traffic light colour used to represent each type of validity or reliability in subsequent tables is a reflection of the mean value of specific FMS related task scores, or subtest scores, as appropriate. A full breakdown of results for each study can be found in S2 Table.

**Table 1. Validity and reliability definitions.**

| COSMIN category | Psychometric Property (if different from COSMIN category) | Definition |
|---|---|---|
| Reliability | Inter-Rater Reliability | The level of agreement between different assessors' scores of children on an assessment tool. |
| | Intra-Rater Reliability | How consistent an assessor is at scoring children using an assessment tool. |
| | Test-retest Reliability | The stability of the children's scores on an assessment tool over a minimum of two time points. |
| | Internal consistency | The level of agreement between items within an assessment tool. |
| Content Validity | | The extent to which an assessment is representative of the components/facets it was designed to measure. |
| Construct Validity | Structural Validity | The degree to which an assessment tool measures what it was designed to measure. |
| | Cross-Cultural Validity | The degree to which an assessment tool and its' normative data can be used to assess FMS in countries other than the one it was designed in. |
| | Hypotheses Testing | The degree to which scores on assessments are consistent with hypotheses made by authors (e.g. internal relationships between subscales, relationships to scores of other assessment tools or differences between relevant groups. |
| Criterion Validity | Concurrent Validity | The level of agreement between two assessment tools. |
| | Predictive Validity | The degree to which performance on an assessment tool can be used to predict performance on another measure, tested at a later date. |

# Results

## Assessment tools

The pre-search identified 33 possible FMS assessment tools of which three were removed for not meeting criteria 1. These were Functional Movement Screen [30, 31], Lifelong Physical Activity Skills Battery [32], New South Wales Schools Physical Activity and Nutrition Survey [33]. Two were removed for failing criteria 3. These were Fundamental Motor Skill Stage Characteristics/ Component Developmental Sequences [34] and the Early Years Movement Skills Checklist [35]. Additionally three tools were identified as being the same assessment tool, with the name translated differently- the FMS assessment tool, the Instrument for the Evaluation of Fundamental Movement Patterns and the Test for Fundamental Movement Skills in Adults [36]. The APM-Inventory [37] and the Passport for Life [38] were removed as

**Table 2. Traffic light system for analysing results of included studies.**

| | Level of Evidence | | | |
|---|---|---|---|---|
| **Statistical Method** | **Poor** | **Moderate** | **Good** | **Excellent** |
| Intraclass Correlation (ICC) [26] | < .5 | .5 - .75 | .75 - .9 | >.9 |
| Pearson Correlation [27] | < .3 | .3 - .6 | .6 - .8 | >.8 |
| Spearman Correlation [27] | < .3 | .3 - .6 | .6 - .8 | >.8 |
| Kappa [28] | < .6 | .6 - .79 | .8 - .9 | >.9 |
| Cronbach's alpha [29] | < .6 | .6 - .7 | .7 - .9 | >.9 |

*NB*: For Kappa statistics, the first three thresholds described by the authors ("none", "minimal" and "weak") were combined to form "poor" in the table above [28]. For Cronbach's alpha, "unacceptable" and "poor" were combined to be classified as "poor" for the purpose of this review [29].

no information could be found explaining the assessment tool, and authors either did not respond to queries, or no contact information could be found for the author. This left 24 assessment tools for inclusion in the systematic review, which reviewed studies if they: (i) used assessment tool(s) identified in the pre-search; (ii) measured validity or reliability quantitatively; (iii) sampled children old enough to be in compulsory education within their country. Studies were not excluded based on sample health or motor competence. Concurrent validity was only examined between the 24 assessment tools identified in the pre-search.

## Included studies

Electronic searches initially identified 3749 articles for review. Fig 1 demonstrates the review process which resulted in 90 studies being selected (for study table see S2 Table).

Included articles explored the validity and/or reliability of sixteen of the assessment tools identified in the pre-search. The search did not identify any articles for the remaining eight assessment tools (see Table 3), so the reliability and validity of these measures could not be evaluated in this review. Only nine of the assessment tools identified in the pre-search assess all three components of FMS: locomotion, object control and balance [24]: the Bruininks-Oseretsky Test of Motor Proficiency (BOT) [40, 41], FMS Polygon [42], Get Skilled Get Active (GSGA) [43], Peabody Developmental Motor Scale (PDMS) (Folio & Fewell, 1983, 2000), PLAYfun [44], PLAYbasic [45], Preschooler Gross Motor Quality Scale (PGMQ) [46], Stay in Step Screening Test [47], and the Teen Risk Screen [48] of which three were product and five were process-oriented. Fig 2 shows a breakdown of the number of assessment tools which

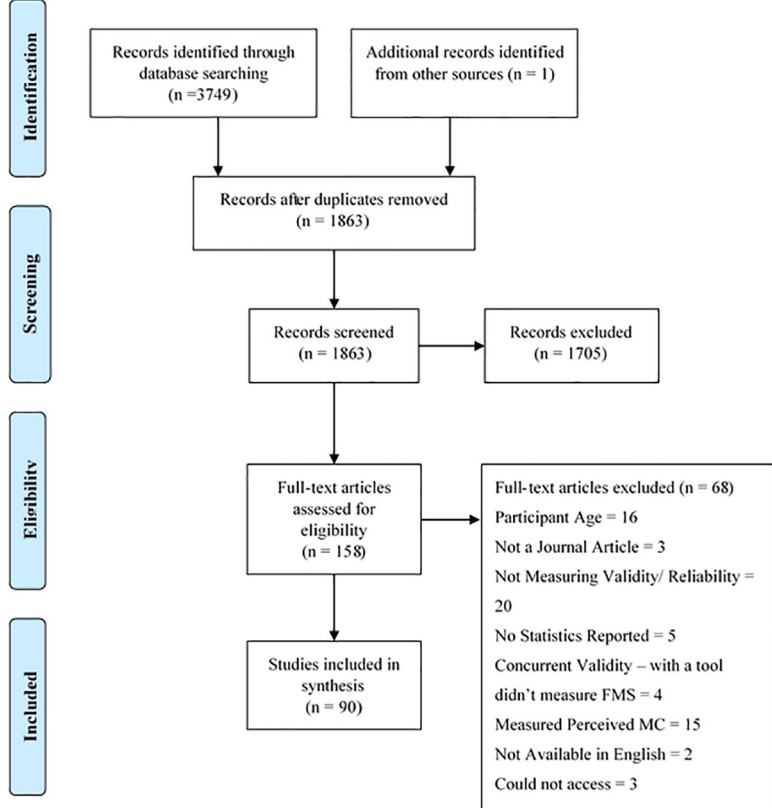

**Fig 1. A PRISMA flow diagram [39] illustrating the review process.**

**Table 3. The psychometric properties measured for each assessment tool found to measure FMS proficiency.**

| Assessment Tool | FMS Measured (subscales) | Outcome(s) | Number of Validity /Reliability Studies | Types of Validity and Reliability Assessed |
|---|---|---|---|---|
| Athletics Skills Track (AST) [a] [98] | AST-1: Crawl, hop, jump, throw, catch, kick, running backwards AST-2: crawl, walk, jump, roll, hopping | Time taken to complete the course | 1 | Test-Retest Reliability Internal consistency |
| Bruininks-Oseretsky Test of Motor Proficiency (BOT) [a] [40, 41] | Balance: static balances (e.g. standing on one leg) and dynamic balance (e.g. walking along a line) Running speed and agility: running, hopping, jumping Upper limb coordination: catching, dribbling, throwing | Time taken to complete tasks, number of tasks completed in a set time limit | 22 | Inter-Rater Reliability Test-Retest Reliability Internal Consistency Structural Validity Concurrent Validity Cross-Cultural Validity Hypothesis testing validity |
| Canadian Agility and Movement Skill Assessment (CAMSA) [a,b] [92] | Jump, slide, catch, skip, hop, kick and run | Time taken to complete the course (converted to points range) and a performance assessment for each skill | 3 | Inter-Rater Reliability Intra-Rater Reliability Test-Retest Reliability Concurrent Validity |
| Children's Motor Skills Protocol (CMSP) [b] [99] | Locomotor: run, broad jump, slide, gallop, leap, hop Object control: overarm throw, underhand roll, kick, catch, stationary strike, stationary dribble | Number of movement characteristics observed for each skill | 0 | N/A |
| Fundamental Motor Skills Test Package (EUROFIT, FMS Test Package) [a] [100, 101] | Balance, jump and run | Time taken to complete 20m shuttle run, time can stand on one leg, and distance jumped | 0 | N/A |
| Fundamental Movement Skill Polygon (FMS Polygon) [a] [42] | Space Covering: Crawling, rolling, running, beam walking, Surmounting Obstacles: skipping, hopping, jumping Object Control: Dribble, throw, catch | Time taken to complete tasks | 1 | Intra-Rater Reliability Structural Validity Concurrent Validity |
| Furtado-Gallagher Computerized Observational Movement Pattern Assessment System (FG-COMPASS) [b] [102] | Locomotor: Hopping, jumping, leaping, skipping, sliding Manipulative: Hitting, catching, kicking, dribbling, throwing | Patterns of movement characteristics for each skill | 1 | Inter-Rater Reliability |
| Get Skilled Get Active (GSGA) [b] [43] | Static balance, jump, run, catch, hop, leap, gallop, kick, skip, hit, throw, dodge | Ability to consistently complete patterns of movements for each skill in a variety of environments/ contexts | 1 | Concurrent Validity |
| Instrument for the Evaluation of Fundamental Movement Patterns [b] [36] | Locomotor: run, jump, gallop, slide, hop Object Control: bounce, catch, kick, strike, throw | Number of points (one per criterion met per skill) | 0 | N/A |

(*Continued*)

**Table 3.** (Continued)

| Assessment Tool | FMS Measured (subscales) | Outcome(s) | Number of Validity /Reliability Studies | Types of Validity and Reliability Assessed |
|---|---|---|---|---|
| Körperkoordinationstest für Kinder (KTK) [a] [103–105] | Walking backwards along beams of varying widths<br>Hopping for height<br>Jumping sideways over a slat<br>Moving sideways on boards | Number of steps walked along the beam, number of successful hops/ jumps/ movements | 10 | Inter-Rater Reliability<br>Structural Validity<br>Concurrent Validity<br>Internal Consistency<br>Hypothesis testing validity |
| Motoriktest für vier- bis sechsjärige Kinder (MOT 4–6) [a] [106] | Gross Motor: jumping, walking, catching, throwing, hopping | Number of jumps completed, time taken to complete tasks etc. Raw scores are converted into a 3 level ranking scale: 0 (not mastered)– 2 (mastered) | 4 | Structural Validity<br>Concurrent Validity<br>Hypothesis testing validity |
| Movement Assessment Battery for Children [a] [107, 108] | Aiming and catching<br>Throwing, catching<br>Balance: static balance (e.g. on one leg), dynamic balance (e.g. walking along the line, jumping, hopping) | Number of successful attempts, length of time balances can be held for | 37 | Inter-Rater Reliability<br>Intra-Rater Reliability<br>Test-Retest Reliability<br>Internal Consistency<br>Predictive Validity<br>Content Validity<br>Structural Validity<br>Cross-Cultural Validity<br>Concurrent Validity<br>Hypothesis testing validity |
| Objectives-Based Motor-Skill Assessment Instrument [b] [109] | run, gallop, hop, skip, jump, leap, slide, strike, bounce, catch, kick, throw | The number of qualitative motor behaviours exhibited across the FMS measured (/45) | 0 | N/A |
| Ohio State University Scale for intra-Gross Motor Assessment (OSU-SIGMA) [b] [110] | Locomotor: walking, running, jumping, hopping, skipping, climbing<br>Object control: throwing, catching, striking, kicking | Levels of development for each skill 1 (least mature)– 4 (mature functional pattern) based on qualitative assessment of movement patterns | 0 | N/A |
| Peabody Developmental Motor Scale (PDMS) [b] [111, 112] | Stationary<br>Locomotion: crawling, walking, running, hopping, jumping<br>Object manipulation: throwing, catching | Score of 0–2 as to the level of skill shown for each FMS (not demonstrated, emerging, proficient | 1 | Concurrent Validity |
| PE Metrics [a,b] [113, 114] | Throwing, catching, dribbling, kicking, striking<br>Hopping, jumping, galloping, sliding, running, skipping | Score of 0–4 for form (how well the movement is executed) and success (the outcome of the movement) | 1 | Structural Validity |
| PLAYbasic [b] [45] | Locomotor: run, hop<br>Throw<br>Kick<br>Balance (dynamic- heel to toe backwards) | Levels of development for each FMS– developing (initial or emerging) or acquired (competent or proficient) | 1 | Inter-Rater Reliability<br>Internal Consistency<br>Concurrent Validity |

(*Continued*)

**Table 3.** (Continued)

| Assessment Tool | FMS Measured (subscales) | Outcome(s) | Number of Validity /Reliability Studies | Types of Validity and Reliability Assessed |
|---|---|---|---|---|
| PLAYfun[b] [45] | Runnings: run a square, run there and back, run, jump and land on two feet<br>Locomotion: skip, gallop, hop, jump<br>Upper body object control: overhand throw, strike, one handed catch, stationary dribble<br>Lower body object control: kick a ball, foot dribble<br>Balance: walk heel-to-toe forwards, walk heel-to-toe backwards, | Levels of development for each FMS– developing (initial or emerging) or acquired (competent or proficient) | 2 | Inter-rater reliability Structural validity Internal Consistency Concurrent Validity Hypothesis Testing Validity |
| Preschooler gross motor quality scale (PGMQ)[b] [46] | Locomotion: Run, jump, hop, slide, gallop, leap<br>Object manipulation: throw, catch, kick, bounce, strike<br>Static balance: one leg balance, tandem one leg balance, walking along the line forwards, walking along the line backwards | Number of qualitative qualities for each FMS each child demonstrates | 0 | N/A |
| Smart Start [b] [115] | Locomotor: run, gallop, hop, leap, jump, slide<br>Object control: strike, bounce, catch, kick, throw | Whether elements of each skill were completed (1 = yes, 0 = no) | 0 | N/A |
| Teen Risk Screen [b] [48] | Posture & Stability (Axial Movement): sitting, standing, bending, stretching, twisting, turning, swinging<br>Posture & Stability (Dynamic Movement): body rolling, starting and stopping, dodging and balance<br>Locomotor Skills (Single Skills): walking, running, leaping, jumping and hopping<br>Locomotor Skills (Combinations): galloping, sliding and skipping<br>Manipulative Skills (Sending Away): carrying, dribbling<br>Manipulative Skills (Maintaining Possession): catching | Extent to which each skill can be performed according to guidelines (0 = cannot perform the skill according to guidelines, 1 = can perform the skill but not according to the guidelines, 2 = can perform the skill) | 1 | Internal Consistency Structural Validity Test-Retest Reliability |
| Test of Gross Motor Development (TGMD)[b] [116–118] | Locomotor: run, gallop, jump, hop, skip, leap, slide<br>Object Control: strike, dribble, catch, kick, throw | The number of qualitative motor behaviours exhibited for each of the FMS measured | 34 | Inter-Rater Reliability Intra-Rater Reliability Test-Retest Reliability Internal Consistency Content Validity Structural Validity Cross-Cultural Validity Concurrent Validity Hypothesis Testing Validity |

(*Continued*)

**Table 3.** (Continued)

| Assessment Tool | FMS Measured (subscales) | Outcome(s) | Number of Validity /Reliability Studies | Types of Validity and Reliability Assessed |
|---|---|---|---|---|
| Victorian Fundamental Movement Skills Assessment Instrument [b] [119] | Catch, kick, run, jump, throw, bounce, leap, dodge, strike | The number of components of each FMS a child has mastered | 1 | Concurrent Validity |
| Stay in Step Screening Test [a] [47] | Static balance (one leg), bounce, catch, hop, run | Duration balance is held for, number of completed throws/catches in a specified timeframe, distance hopped, time taken to complete task (e.g. 50m run) | 0 | N/A |

NB: [a] = product-oriented, [b] = process-oriented

measure each aspect of FMS. Other aspects of motor development (e.g. the MABC has a manual dexterity subscale) were measures by the included assessment tools, but this review specifically focused on FMS.

## Participants

The included studies recruited a total of 51,408 participants aged between three and seventeen years of age, with sample sizes that ranged from 9 to 5210 (mean = 556 [SD = 1000] median = 153 [IQR = 652]). Twenty-four studies included additional sample demographics, with seven studies recruiting children with movement difficulties [49, 50], Cerebral Palsy [51, 52] or Developmental Coordination Disorder [53–55]. Two studies included participants with Autistic Spectrum Disorder [56, 57], and another study recruited children from special educational needs (SEN) schools [58]. Eight defined themselves as sampling children with learning and/or attentional problems [54, 59–65], three studies recruited children with visual impairments [66–68], and the sample of one study included children with a disability or chronic health condition [69]. Information regarding socioeconomic status (SES) was included in one article which stated they sampled from low SES [70], while two studies recruited samples from indigenous populations (in Australia and Canada, respectively) [44, 71], the latter of which focused on the recruitment of children whose mothers drank alcohol during pregnancy [71].

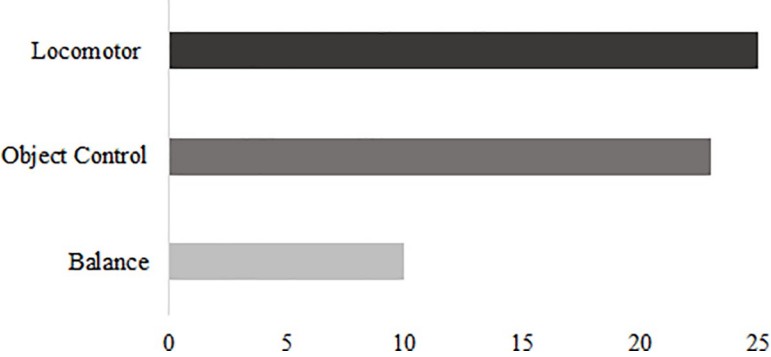

**Fig 2. Graphical representation of the number of assessment tools which evaluate each of the three aspects of FMS.**

Studies evaluating the validity and reliability of FMS assessment tools were conducted in 29 countries, with Australia hosting the most studies (13) [50, 56, 71–77], followed by Brazil (12 studies) [53, 57, 66, 70, 78–85] and the USA (nine studies). Eight studies were carried out in Belgium [49, 58, 63, 86–89] and seven in Canada [43, 54, 60, 90–94]. The remaining 23 countries spanned Europe (23 studies from 15 countries), Asia (10 studies from 7 countries), South America (one study from Chile) and Africa (one study conducted in South Africa). Two studies did not provide any information regarding where the sample was recruited from [95, 96].

## COSMIN quality assessment

Fig 3 shows the results of the generalisability subscale of the quality assessment for the included studies. The COSMIN checklist [25] revealed multiple issues with reporting in the included studies, with 85% of studies not providing enough information to make a judgement about missing responses, and 76% of studies failing to report the language with which the assessment tool was conducted. Additionally, over a third of the studies included in this review did not adequately describe the method of recruiting participants, the age of participants, or the setting in which testing was conducted.

## Assessment tool categorisation

Observational assessment methods were defined categorically as either assessing FMS using a "process" or "product-oriented" methodology [97]. Process-oriented measures require decisions to be made as to whether children are meeting specific performance criteria whilst completing skills (e.g. when running, is the non-support leg is bent at a ninety degree angle?). Product-oriented assessments focus on the outcome of movements (e.g. how quickly can a child can complete a movement?). Given these two different approaches to measuring FMS, which can used for different purposes in the literature, they were distinguished for this review. Of the 24 assessment tools identified, nine were product-oriented, thirteen were process-oriented, and two assessment tools included both process and product methodologies (see Table 3).

## Product oriented assessments

Despite the pre-search identifying nine product-oriented assessments in the FMS literature, the systematic review only identified research on the validity and reliability of six of these measures (described below). No evaluations of the psychometric properties of any of the following assessments were found: the APM inventory [37], the FMS Test Package [100, 101] and the Stay in Step Screening Test [47].

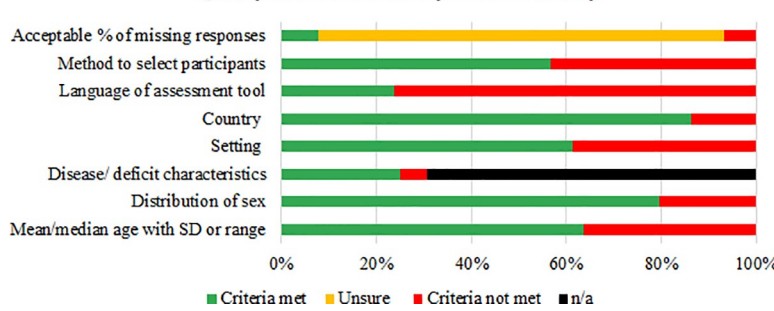

**Fig 3. Summary of the generalisability subscale of the COSMIN checklist.**

**Table 4. Reliability and validity of the MABC.**

| Study | | Reliability | | | | Validity |
|---|---|---|---|---|---|---|
| | | IeR | IaR | TR | IC | Pr |
| Chow et al. [121] | MABC | green | | orange | | |
| Croce et al. [123] | | | | green | | |
| Ellinoudis et al. [124] | | | | | orange | |
| Smits-Engelsman et al. [49] | | green | | | | |
| Bakke et al. [66] | MABC-2 | green | | green | yellow | |
| Borremans et al. [57] | | | | | orange | |
| Darsaklis et al. [96] | | green | | | | |
| Holm et al. [120] | | orange | red | | | |
| Hua et al. [122] | | green | | green | red | |
| Jaikaew et al. [125] | | green | | | | |
| Kita et al. [126] | | | | | orange | |
| Valentini et al. [83] | | green | yellow | | yellow | yellow |
| Wuang et al. [55] | | | | green | yellow | |

*NB*: IeR = interrater IaR = intra rater, TR = test-retest, IC = internal consistency, St = Structural, Ct = content, Pr = predictive. ■ = poor (ICC < .5, $r < .3$, $\kappa < .6$, $\alpha < .6$), ■ = moderate (ICC = .5 - .75, $r = .3$ - .6, $\kappa = .6$ - .79, $\alpha = .6$ - .7), ■ = good, (ICC = .75 - .9, $r = .6$ - .8, $\kappa = .8$ - .9, $\alpha = .7$ - .9) ■ = excellent validity/reliability (ICC > .9, $r > 8$, $\kappa > .9$, $\alpha > .9$).

**Movement Assessment Battery for Children (MABC).** Twenty-three studies evaluated the validity and/or reliability of the MABC or MABC-2. All of the ten COSMIN categories this review focused on (see Table 1), were evaluated for the MABC. Overall there was strong evidence for inter-rater reliability for these assessments (Table 4). However, there were more mixed results for other aspects of validity and reliability, with the weakest evidence being found in support for internal consistency. Intra-rater reliability was only looked at in two studies [83, 120] with poor intra-rater reliability (ICC = .49 for both the balance and aiming and catching subtest) demonstrated in the study exploring this construct in Norwegian children [120]. There was good evidence for test-retest reliability, with only one out of five studies in a sample of teenagers [121] finding moderate correlations (mean ICC for FMS skills = .74). An adapted version of the MABC-2 was also tested (e.g. increasing the colour contrast on the ball), with results showing that the modified version was a reliable assessment tool for use with children with low vision (inter-rater reliability–ICC = .97; test-retest reliability–ICC = .96; internal consistency- Cronbach's alpha ranged from 0.790 to 0.868) [66]. Strong evidence for content validity was found for both the Brazilian [83] and the Chinese [122] versions of the assessment tool, with concordance rates amongst experts ranging from 71.8%-99.2%. Additionally, one study found that children with Asperger syndrome perform worse on all three subtests of the MABC than typically developing children, as hypothesised [57].

Cross-cultural validity was studied in four papers, looking at Swedish, Spanish, Italian, Dutch and Japanese samples in comparison to US or UK norms [88, 127–129]. Results showed that UK norms were not suitable for use to evaluate the performance of Italian children, as significant differences were found for eleven of the twenty seven items on the MABC-2 [129]. Differences were also found between the performance of UK children and Dutch children, however these differences were not statistically significant. The US standardised sample was found to be valid for a Swedish sample [127], but not for a Spanish sample, for which US norms left a large proportion of the sample below the 15th percentile [128].

Structural validity was assessed by ten studies, with six finding evidence for a three factor (manual dexterity, aiming & catching and balance) model [78, 122, 126, 129–131]. One study

found a four factor solution, with a general factor for age band 1, four factors with balance split into static and dynamic for age band 2, and a 3 factor correlated model for age band 3 [132]. Similarly, another study found evidence for a bifactor model with one general factor, and three sub-factors for age band one [81]. Evidence was also found for a five factor solution, with balance and manual dexterity each split into two factors [124]. An adolescent study found a two factor model (manual dexterity and aiming and catching) was more appropriate as ceiling effects were evident on balance tasks [133].

The results of the COSMIN quality assessment of MABC studies show that two studies which found excellent results, had the lowest quality ratings, in which they met 13% and 29% of generalisability and inter-rater reliability criteria respectively [96, 125]. Additionally, the singular study which found MABC normative data to be valid in another country only had a quality rating of 39% [127]. The MABC study with the best quality rating (81% of criteria met), only found moderate results for internal consistency [126], and the single study which found that MABC norms data are cross-culturally valid, only had a quality rating of 39%. When considering COSMIN quality ratings alongside the results of these studies, it would suggest that caution should be taken when interpreting the results of studies exploring the psychometric properties of the MABC.

**Bruininks-Oseretsky Test of Motor Proficiency (BOT).** Twelve studies stated that they explored the validity and reliability of the BOT, BOT-2 or BOT-2 Short Form (SF), of which six reported results that could be quantified into poor, moderate, good and excellent evidence, which are detailed in Table 5. Three studies looked at the inter-rater reliability of the BOT, all of which found good evidence in support of this aspect of reliability [54, 71, 96], however one of these studies provided no information about the sample, including size and demographic information [96]. The results for test-retest reliability were more mixed than for the MABC, with the two studies finding low correlations on scores between tests sampling from children with Cerebral Palsy (ICC = .4) [52] and children living in aboriginal communities in Australia (mean ICC for FMS = .097) [71]. One study did show evidence of the BOT being a reliable measure of FMS in children with intellectual deficits [65]. One study explored the cross-cultural validity of the BOT-2 norm scores with a large Brazilian sample (n = 931) and found mixed results [79]. Results showed that Brazilian children outperformed the BOT normative data on bilateral coordination, balance, upper-limb coordination, and running speed and agility subtests, but similar percentile curves were found for both populations on upper limb coordination and balance subtests [79].

Five studies explored the structural validity of the BOT. The BOT-2 SF was also found to have good structural validity once mis-fitting items were removed for children aged 6–8 years, but ceiling effects were found for older children (aged 9–11 years)[134]. Two studies exploring

**Table 5. Validity and reliability of the BOT.**

| Study | | IeR | IaR | TR | IC | Pr |
|---|---|---|---|---|---|---|
| | | | | **Reliability** | | **Validity** |
| Iatridou & Dionyssiotis [51] | **BOT** | | | 🟩 | | |
| Liao et al. [52] | | | | 🟥 | | |
| Wilson et al. [54] | | 🟩 | | | | |
| Darsaklis et al. [96] | **BOT-2** | 🟩 | | | | |
| Wuang & Su [65] | | | | 🟩 | 🟩 | |
| Lucas et al. [71] | **BOT-2 SF** | 🟨 | | 🟧 | | |

*NB*: IeR = interrater IaR = intra rater, TR = test-retest, IC = internal consistency, St = Structural, Ct = content, Pr = predictive. 🟥 = poor (ICC < .5, $r$ < .3, $\kappa$ < .6, $\alpha$ < .6), 🟧 = moderate (ICC = .5 -.75, $r$ = .3 - .6, $\kappa$ = .6 - .79, $\alpha$ = .6 - .7), 🟨 = good, (ICC = .75 -.9, $r$ = .6 - .8, $\kappa$ = .8 - .9, $\alpha$ = .7 - .9) 🟩 = excellent validity/reliability (ICC >.9, $r$ > 8, $\kappa$ >.9, $\alpha$ > .9)

structural validity found good evidence utilising Rasch analysis, with results indicative of unidimensionality, with the overarching factor accounting for 99.8% [64] and 82.9% [73] of the variance in test scores for children with intellectual deficits (BOT), and typically developing children (BOT-BF), respectively. Similarly to the results of the Rasch studies, one additional study found that the four subscales were correlated, so a bifactor model, with an overarching motor skill factor, and four correlated sub-factors [81]. When the subscales and composite scales were evaluated separately using Rasch analysis, one study found multiple issues with fine motor integration, bilateral coordination, balance and body coordination which limit the justification of their use including multi-dimensional scales, items working differently for males and females, disordered item difficulty ratings, and/or the ability of the subscale/ composite score to differentiate between abilities [135].

The quality of the studies evaluating the validity and reliability of the BOT may have influenced the results though, as the study with the greatest quality rating (83%) found good results for inter-rater reliability [71], but two studies with lower ratings (13% [96] and 53% [54]) reported excellent results for this psychometric property, suggesting that reliability scores may have been inflated by poorer quality studies. Additionally, the reviewed BOT studies only evaluated seven of the ten COSMIN categories (see Table 3).

**Other product-oriented assessment tools.** Three studies evaluated the validity and reliability of the Körperkoordinationstest für Kinder (KTK) [77, 80, 136]. Two studies looked at the structural validity of the KTK, and found adequate evidence to support a one factor structure, interpreted as representing "body coordination" [77, 80]. The internal consistency of the KTK was consistently found to be good across samples in Finland, Portugal and Belgium (α ranged from .78 - .83), however, as hypothesised there were significant differences between groups, in which children from Portugal and Belgium performed worse than Finnish participants [136]. Additionally, there was evidence of high inter-rater reliability (94% agreement) [77].

Two studies evaluated the validity and reliability of the Athletic Skills Track (AST) [98, 137]. The results of both studies suggest that the AST has good test-retest reliability with intraclass correlations ranging from .8 [137] to .88 [98]. Cronbach's alpha was used in one of these studies to examine internal consistency, with results ranging from .7-.76 for the three versions of the AST [137]. It is, however, important to note that only two psychometric properties from the COSMIN checklist [25] were evaluated, and the quality ratings for these studies were lower than 60%. The psychometric properties of the FMS Polygon were tested in one study [138], finding strong evidence for intra-rater reliability (ICC = .98). Factor analysis also explored the structure of the assessment tool, revealing four factors: object control (tossing and catching a volleyball), surmounting obstacles (running across obstacles), resistance overcoming obstacles (carrying a medicine ball) and space covering skills (straight running). These psychometric properties of the FMS Polygon, should however, be interpreted with caution, as the above study only had a quality rating of 43% [138].

The structural validity of the MOT 4–6 was evaluated by one study with a high quality rating (79%) using Rasch analysis, which established four of the items had disordered thresholds and needed to be removed from the assessment (grasping a tissue with a toe, catching a tennis ring, rolling sideways over the floor and twist jump in/out of a hoop). Results also showed that with one additional item removed (jumping on one leg into a hoop), there was an acceptable global model fit for the MOT 4–6 [139].

## Process-oriented assessments

Thirteen process-oriented assessment tools were identified by the pre-search as measuring FMS. Of these, seven had been evaluated for validity and reliability (described below). No

research was found evaluating the psychometric properties of the: Children's Motor Skills Protocol (CMSP)[99], Instrument for the Evaluation of Fundamental Movement Patterns [36], Objectives-Based Motor-Skill Assessment Instrument [109], Ohio State University Scale for intra-Gross Motor Assessment (OSU-SIGMA) [110], Preschooler Gross Motor Quality Scale (PGMQ) [46] and Smart Start [115].

**Test of Gross Motor Development (TGMD).**   The results of twenty-one studies which evaluated the psychometric properties of various versions of TGMD can be found in Table 6. Nine out of ten COSMIN psychometric properties were evaluated by TGMD studies. Consistently good evidence for inter-rater and intra-rater reliability was observed, with only one study finding less than 'good' (moderate) correlations when testing sessions were video recorded [140]. One study evaluated these aspects of reliability using a Content Validity Index (CVI) and found good evidence for both inter and intra-rater reliability when testing Chilean children, with CVIs ranging from .86 to .91 [141]. An additional study evaluated the inter and intra-rater reliability of the TGMD second and third editions using percentage agreement [69]. Results showed agreement for inter-rater reliability was 88% and 87% for the TGMD-2 and TGMD-3 respectively, and for intra-rater reliability the percentage agreement was 98% for the TGMD-2 and 95% for the TGMD-3 [69]. Fewer studies examined the test-retest reliability of the TGMD, but those that did demonstrated that for the TGMD-2 [63, 68, 82, 142, 143], a short version of the TGMD-2 modified for Brazilian children [84] and the TGMD-3 [56, 85, 144, 145] participants score similarly when they are tested on multiple occasions. Strong test-

**Table 6. Validity and reliability of the TGMD.**

| Study | | Reliability | | | | Validity |
|---|---|---|---|---|---|---|
| | | IeR | IaR | TR | IC | Pr |
| Allen et al. [56] | TGMD-2 | | | | green | |
| Barnett et al. [72] | | green | | | | |
| Capio et al. [59] | | green | green | | yellow | |
| Garn & Webster [147] | | | | | yellow | |
| Houwen et al. [68] | | yellow | green | green | orange | |
| Issartel et al. [142] | | | | yellow | | |
| Kim et al. [143] | | green | | green | yellow | |
| Lopes et al. [146] | | yellow | | | | |
| Simons et al. [63] | | | | green | green | |
| Valentini et al. [82] | | yellow | | green | green | |
| Ward et al. [148] | | yellow | | | | |
| Valentini et al. [84] | TGMD-2 SF | green | green | green | orange | |
| Allen et al. [56] | TGMD-3 | green | green | green | yellow | |
| Brian et al. [67] | | green | | | | |
| Estevan et al. [149] | | green | green | | yellow | |
| Maeng et al. [150] | | green | green | | | |
| Magistro et al. [151] | | green | | green | | |
| Rintala et al. [140] | | orange | orange | | | |
| Valentini et al. [85] | | green | yellow | green | orange | |
| Wagner et al. [144] | | green | | green | green | |
| Webster & Ulrich [145] | | | | green | green | |

*NB*: IeR = interrater IaR = intra rater, TR = test-retest, IC = internal consistency, St = Structural, Ct = content, Pr = predictive. ■ = poor (ICC < .5, $r < .3$, $\kappa < .6$, $\alpha <$ .6), ■ = moderate (ICC = .5 -.75, $r = .3 - .6$, $\kappa = .6 - .79$, $\alpha = .6 - .7$), ■ = good, (ICC = .75 -.9, $r = .6 - .8$, $\kappa = .8 - .9$, $\alpha = .7 - .9$) ■ = excellent validity/reliability (ICC >.9, $r > 8$, $\kappa >.9$, $\alpha > .9$)

retest reliability was evidenced with a CVI of .88 [141] and Bland Altmann plots found 95% confidence intervals were within one standard deviation [77], with .96 agreement ratio [146]. Evidence for internal consistency was more mixed, but there was strong evidence that all items in the TGMD-3, once modified for children with ASD and visual impairments could still measure FMS as an overarching construct [56, 67]. Evidence for good internal consistency of the TGMD was also found when testing children with intellectual deficits [59].

Sixteen studies evaluated the structure of the items within various editions of the TGMD, consistently finding a two factor model (locomotion and object control) for the TGMD [152], TGMD-2 [59, 63, 68, 77, 82, 142, 143, 146, 147], TGMD-2 SF [84] and TGMD-3 [85, 144, 145, 149, 151], as predicted by multiple studies [59, 146, 149, 152]. It is, however, important to note that some of these models enabled cross-loading of items [e.g. 147], some models were hierarchical in nature [77] and in one case a two factor model, whilst best fit, explained only 50% of the total variance [142]. Evidence was however found to suggest that the structural validity of the TGMD is stable across countries, with the data from populations in Greece, Brazil, Germany, the USA, South Korea and Portugal all evidencing a two factor model [67, 82, 143, 144, 146, 152].

The content validity of the Brazilian translation of the TGMD-2 and TGMD-3 was evaluated by two studies, with stronger evidence for the validity of the TGMD-2 (CVI = .93 for clarity and .91 for pertinence) than the TGMD-3 for which the CVI for the clarity of the instructions only reached .78 [82, 85]. The Spanish translation of the TGMD-2 was also tested for clarity and pertinence, with results finding a CVI of .83 [141]. Cross cultural validity was investigated in one study that compared Flemish children with intellectual deficits to US normative data [63], which found significant differences, with large effect sizes (1.22–1.57), indicating US standardised data was inappropriate for use as a comparison within this population. Additionally, a large study based in Belgium hypothesised that Belgian children would perform similarly to US norms on locomotor scores, but that Belgian children would score lower on object control tasks, however, Belgian children had significantly worse GMQ, locomotor and object control scores, thus showing that US normative data was not appropriate for this sample [153]. The COSMIN quality rating of TGMD studies did not appear to effect results, as the relative quality ratings of all studies that found excellent results only varied by 16% [56, 59, 61, 63, 68, 72, 82, 84, 85, 144] (54–70%). However, predictive validity was not explored by the included TGMD studies.

**Other process-oriented assessment tools.** The psychometric properties of the FG-Compass [102] were evaluated in one study, in which expert scores were compared to undergraduate student scores [154]. Results showed kappa values ranging from .51-.89, with moderate levels of agreement on average ($m$ = .71). PLAYbasic was found to have good inter-rater reliability (mean ICC = .86), and moderate internal consistency (mean $\alpha$ = .605) in one study [44]. Two studies evaluated PLAYfun, finding good to excellent inter-rater reliability (ICC ranged from .78 - .98) and good internal consistency (average $\alpha$ = .78) [44, 91]. Additionally, hypotheses testing validity and structural validity were assessed, with performance increasing with age as hypothesised, and an acceptable model fit for the proposed five factor structure [91]. Despite the quality ratings of these studies varying, (43% and 76%), the higher quality study found the more promising results [91]. One study evaluated the psychometric properties of the Teen Risk Screen [48], with results demonstrating good evidence for the internal consistency (mean $\alpha$ = .75) and test-retest reliability (mean $r$ = .64) of subscales. Confirmatory factor analysis (CFA) was used to evaluate the structural validity of the Teen Risk Screen, however, the analysis was not completed on the model they proposed (6 subscales). Authors claimed that due to small sample sizes, only three of the six subscales were evaluated separately, and the final three were grouped together. As this analysis did not measure the intended model, results

are not detailed in this review. Get Skilled Get Active (GSGA), the Peabody Developmental Motor Scales (PDMS-2) and the Victorian FMS assessment were all used in concurrent validity studies, however, no articles were found evaluating any other aspects of validity and reliability of these measures.

## Combined assessments

Two assessment tools from the pre-search measure both product- and process-orientated aspects of movement: Canadian Agility and Movement Skill Assessment (CAMSA) [92] and PE Metrics [113, 114]. There is limited evidence for the reliability of the CAMSA with one study finding moderate effect sizes for inter-rater, intra-rater and test-retest reliability, as well as internal consistency [92]. One other study found strong evidence for the test-retest reliability of the CAMSA [74], however that study had a lower quality rating (49% compared to 77%). One study evaluated the structural validity of PE Metrics using Rasch analysis and found good evidence that all of the items were measuring the same overarching set of motor skills [155]. It is, however, necessary to interpret this result with caution, as the COSMIN quality rating for this study was only 43%.

## Concurrent validity

Limited evidence was found for concurrent validity across the 23 assessment tools included in the review (see Table 7). A large proportion of the studies exploring this aspect of validity did so against either the MABC (15 studies) or the TGMD (10 studies).

**Between product-oriented.** The findings of studies exploring the concurrent validity of product-oriented assessment tools mostly yielded good results, with only three out of thirteen studies finding less than good evidence for correlations between measures. Of these three studies, one found a poor correlation (kappa = .43) between the MABC and the BOT [60], and two studies found moderate correlations between the MABC and the short form of the BOT [93], as well the AST and the KTK, as hypothesised [137]. Two studies evaluated the concurrent validity of the BOT-2 complete form, and the BOT-2 short form [62, 156]. One found poor correlations between subtests ($r$ ranged from .08 - .45) [156], and the other reported moderate correlations between tasks in a sample of children with ADHD ($r$ ranged from .12 - .98) [62]. A modified version of the KTK (with hopping for height removed) was also compared to the standard KTK, which was found to have high levels of validity [89]. One study used Pearson

**Table 7. Concurrent validity of assessment tools.**

| | | Product-Oriented | | | | | | | Process-Oriented | | |
| | | AST | BOT | | KTK | MOT 4–6 | MABC | | | FMS Polygon | GSGA | PDMS | TGMD |
|---|---|---|---|---|---|---|---|---|---|---|---|---|---|
| Product-Oriented | AST | | | | | | | | | | | | |
| | BOT | | 1 | 1 | | | | | | | | | |
| | KTK | 1 | 1 | 1 | | 1 | | | | | | | |
| | MOT 4–6 | | | | 1 | | | | | | | | |
| | MABC | | 1 | 1 | 3 | 1 | | | | | | | |
| | FMS Polygon | | | | | | | | | | | | |
| Process-Oriented | GSGA | | | | | | | | | | | | |
| | PDMS | | | | | 1 | | | | | | | |
| | TGMD | | | | 1 | | 2 | 2 | 1 | 1 | 1 | | 2 |

*NB*: ■ = poor (ICC < .5, $r$ < .3, $\kappa$ < .6, $\alpha$ < .6), ■ = moderate (ICC = .5 -.75, $r$ = .3 - .6, $\kappa$ = .6 - .79, $\alpha$ = .6 - .7), ■ = good, (ICC = .75 -.9, $r$ = .6 - .8, $\kappa$ = .8 - .9, $\alpha$ = .7 - .9) ■ = excellent validity/reliability (ICC >.9, $r$ > 8, $\kappa$ >.9, $\alpha$ > .9)

correlations to evaluate the concurrent validity between the MOT 4–6 and the KTK, with results showing moderate correlations for children aged 5–6 (mean $r$ = .63), as was hypothesised prior to testing ($r >$.6). In addition to the results detailed in Table 6, one study looked at the concurrent validity of assessing children using the MABC in person and via tele-rehabilitation software, with results showing no significant difference between scores, as hypothesised [76]. As well as this, the MABC and the BOT-SF had a positive predictive value of .88, with twenty one out of twenty four children testing positively for motor coordination problems also scoring below the fifteenth percentile on the MABC [90].

**Between process-oriented.** One study utilised the TGMD to explore the concurrent validity of the GSGA assessment tool [97]. Significant differences were found between the number of children who were classified as mastering FMS versus those who had not, in which GSGA was more sensitive and classified a greater number of children as exhibiting non-mastery [97]. Three studies also explored the relationship between multiple versions of the TGMD. Results revealed that children with ASD perform better on the TGMD-3 with visual aids compared to the standard assessments [56]. Similarly, modified versions of the TGMD-2 and TGMD-3 were both found to be valid for use in children with visual deficits [67]. Additionally, one study showed significant differences between subtest scores on the second and third editions of the TGMD across year groups and gender, in which participants performed better on the TGMD-2 [69].

**Between product- and process-orientated.** The results comparing process and product-oriented assessment tools against each other were also mixed, particularly with regards to the concurrent validity between the MABC and the TGMD, for which correlations ranged from .27-.65 [53, 68, 82, 83, 157]. Study quality did not appear to have an effect on the size of the correlation between the MABC and the TGMD. Two studies also reported significant differences in level of agreement on percentile ranks [53, 157]. The KTK and the TGMD-2 also differed significantly in terms of their classifications of children into percentile ranks [70]. The concurrent validity of the CAMSA and both the PLAYbasic and PLAYfun assessment tools were assessed by one study, which found moderate correlations between CAMSA and both PLAY assessment tools, smaller than was hypothesised [44]. Lastly, good cross-product/process concurrent validity was reported between the MABC and the PDMS [122], as well as the CAMSA and the Victorian FMS Assessment Tool [74] and the TGMD and the FMS Polygon, as hypothesised [138].

## Discussion

The aim of the review was to evaluate the psychometric properties of observational FMS assessment tools for school-age children. There were no studies evaluating the validity or reliability of eight (33%) of the available measures (from 24 identified tools). Of the remaining sixteen, nine (38%) assessment tools only had a single study examining their psychometric properties. Multiple papers evaluating various aspects of validity and reliability were only found for the: MABC (37studies), TGMD (35 studies), BOT (22 studies), KTK (10 studies), CAMSA (3 studies), the MOT 4–6 (4 studies) and PLAYfun (2 studies).

The TGMD was the assessment tool with the most consistently positive evidence in favour of validity and reliability. However, it is important to consider the suitability of observational assessment tools for use in schools, alongside the evidence for the psychometric properties of measures [158]. Recent research by Klingberg et al. established a framework to evaluate the feasibility of implementing FMS assessments in schools [22]. One of the criteria for feasibility detailed in the report was the type of assessment, in which it was stated that product-oriented measures were preferable because they require less training, and are less prone to error. So

despite the TGMD being the assessment tool with the greatest evidence for validity and reliability, it is arguably less feasible to implement in schools settings because it is process-orientated [22]. Notably, despite the strong evidence for the psychometric properties of the TGMD, this assessment tool does not measure balance. Recent research has established that balance is an important aspect of FMS [24] so it is important to recognise the limitations of using tools which do not measure such skills. It seems reasonable to suggest that exploration of the FMS proficiency of children in schools should involve an assessment tool which encompasses loco-motor skill, object control and balance to enable insights into the skills which underpin a child's ability to participate in physical activity [5].

The systematic review found nine product-oriented assessment tools. The product-oriented measure with the most promising feasibility in Klingberg et al.'s review [22], which was also included in this review, was the AST [98]. There is, however, insufficient evidence on the psychometric properties of this assessment tool to allow confidence in its use, as only two of the ten forms of validity and reliability specified by the COSMIN checklist [25] were evaluated in the studies we reviewed [98, 137]. Moreover, the AST assesses how quickly a child can perform a range of FMS, rather than how well each child can perform these movements, arguably limiting the value of the results obtained by the assessment because it focuses solely on speed of movement. Additionally, this assessment, again, does not include a measure of balance. Thus, it would also not provide a school with a comprehensive picture of pupils' FMS.

Only three of the product-oriented assessment tools in this review measure locomotion, object control and balance. The measure with the largest number of psychometric properties evaluated from these three tools was the MABC. However, the evidence for the validity and reliability of this assessment tool was very mixed, and the quality of the studies that found strong evidence for its psychometric properties was questionable. Moreover, the MABC requires specialist equipment such as mats, which contribute to making the measure expensive to buy (approximately £1000). This may not be feasible with increasing pressure on school budgets [159]. The MABC also takes an extended period of time to administer (30–60 minutes), and must be delivered 1-to-1 by a trained professional. These time and resource constraints makes it difficult to recommend to schools as a feasible screening measure, despite it being advocated as the current 'gold standard' for detecting motor skill deficits in Europe [160].

The BOT was the next most explored product-oriented assessment tool that measures all three aspects of FMS, and whilst it was not considered in the Klingberg et al. evaluation of the feasibility of assessments [22] it is again, notably costly to purchase and takes between 45–60 minutes to assess each child. Thus, with teachers feeling increasingly concerned about the time they have available to cover the 'core' assessed curriculum [161], it appears unlikely that schools would be willing to invest the time required to universally assess FMS all pupils using this tool. The final product-oriented assessment tool which assesses all three aspects of FMS is 'Stay in Step' [47]. There were, however, no studies found that evaluate the psychometric properties of this assessment tool. This is particularly problematic as it is already being used within schools in Australia. It is crucial that assessment tools are developed using a rigorous process which ensures they have strong psychometric properties. Schools have limited capacity for new initiatives, so it is important that assessment tools being marketed to them are not only feasible for use, but can also accurately measure FMS and identify children that need additional support, otherwise the assessment becomes redundant, and a waste of already stretched resources. In summary, this review offers a guide to help researchers, clinicians and teachers make an informed decision on available observational FMS assessment tools. However, as discussed, there are a number of limitations with regard to all available assessments which need to be considered. There is an appetite amongst health practitioners to use schools as settings

for motor skill assessments [19] but currently available measures have limited utility within such environments. The majority of existing assessments are commercial products creating significant financial implications for schools that wish to deploy these tests at scale. Moreover, a lot of these tests require a substantial investment of time as they are designed to be conducted with a single child, with children tested in a serial manner. Meanwhile, the tests that do exist without some of these limitations (e.g. AST and KTK) have limited evidence for their validity and reliability, and/or do not measure all three aspects of FMS [24], which limits the justification of their use within evidence-based health and educational practice. Either, assessment tools with strong evidence for validity and reliability (e.g. TGMD) need to be modified to be feasible for use in schools, or feasible tests (e.g. AST) need more research to be done to establish psychometric properties. Currently, schools would have to choose an assessment tool based on either feasibility or strong psychometric evidence alone, however, it is known from educational research that there needs to be a trade-off between the two for school-based initiatives to be implemented consistently, and effective [158].

This review reveals that there are a large number of novel observational assessment tools that have been and are continuing to be developed to measure FMS proficiency in school-age children. We would argue that authors must consider from the outset how to make such tools feasible for use in schools. The results also showed that not enough FMS assessment tools being developed include all three aspects of FMS. In particular, balance has been neglected despite research establishing it as a crucial addition to this group of motor skills [24]. In addition, it is important that the evaluation of the psychometric properties of these new tools is comprehensive, spanning all psychometric properties outlined by the COSMIN guidelines [25]. One of the main limitations of the studies included in this review was the tendency for the authors to be selective about which aspects of validity and reliability were tested. All aspects of validity/ reliability in the COSMIN guidelines evaluated by this review were measured by at least one study, however, no single aspect was measured than more by half of the studies. The most commonly measured aspects of validity and reliability were inter-rater reliability (45% of studies) and structural validity (42% of studies). Future research should consider evaluating predictive validity (1% of studies) and cross-cultural validity (7% of studies) using normative data more often, as these were the most neglected psychometric properties. The lack of consistency for measuring psychometric properties makes it difficult to draw any conclusions about the quality of the tools advertised, particularly when the reports involve the testing of specially selected samples (e.g. children with ASD) where there are fewer studies undertaken.

## Conclusion

It is clear from the published literature there is insufficient evidence to justify the use of current FMS assessment tools for screening in schools. It follows that: (i) researchers, teachers, and clinicians should be cautious when selecting existing measures of FMS for use in these settings; (ii) there is a need to develop low cost, reliable and valid measures of FMS that are suitable for testing large numbers of children within school settings.

## Supporting information

**S1 Checklist. PRISMA 2009 checklist.**
(DOC)

**S1 Table. Search strategy.**
(DOCX)

**S2 Table. Study table.**
(DOCX)

## Author Contributions

**Conceptualization:** Lucy H. Eddy, Daniel D. Bingham, Mark Mon-Williams, Liam J. B. Hill.

**Data curation:** Lucy H. Eddy, Kirsty L. Crossley, Nishaat F. Shahid, Marsha Ellingham-Khan, Ava Otteslev, Natalie S. Figueredo.

**Formal analysis:** Lucy H. Eddy, Ava Otteslev.

**Investigation:** Lucy H. Eddy, Daniel D. Bingham, Kirsty L. Crossley, Nishaat F. Shahid, Marsha Ellingham-Khan, Ava Otteslev, Natalie S. Figueredo.

**Methodology:** Lucy H. Eddy, Daniel D. Bingham, Mark Mon-Williams, Liam J. B. Hill.

**Project administration:** Lucy H. Eddy.

**Supervision:** Daniel D. Bingham, Mark Mon-Williams, Liam J. B. Hill.

**Validation:** Daniel D. Bingham, Mark Mon-Williams, Liam J. B. Hill.

**Visualization:** Lucy H. Eddy.

**Writing – original draft:** Lucy H. Eddy.

**Writing – review & editing:** Lucy H. Eddy, Daniel D. Bingham, Marsha Ellingham-Khan, Ava Otteslev, Natalie S. Figueredo, Mark Mon-Williams, Liam J. B. Hill.

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
