## [Decision Letter · Decision Letter 0]

18 May 2020

PONE-D-20-03384

The validity and reliability of observational assessment tools available to measure fundamental movement skills in school-age children: a systematic review.

PLOS ONE

Dear Dr. Eddy

Thank you for submitting your manuscript to PLOS ONE. After careful consideration, we feel that it has merit but does not fully meet PLOS ONE’s publication criteria as it currently stands. Therefore, we invite you to submit a revised version of the manuscript that addresses the points raised during the review process.

We would appreciate receiving your revised manuscript by Jul 02 2020 11:59PM. To enhance the reproducibility of your results, we recommend that if applicable you deposit your laboratory protocols in protocols.io, where a protocol can be assigned its own identifier (DOI) such that it can be cited independently in the future. For instructions see: http://journals.plos.org/plosone/s/submission-guidelines#loc-laboratory-protocols

We look forward to receiving your revised manuscript.

Kind regards,

Ali Montazeri

Academic Editor

PLOS ONE

Journal Requirements:

2. We noticed that the search of your systematic review was last performed in January 2019. Please ensure that the search is up to date and that your systematic review includes any new studies published since then

Reviewers' comments:

Reviewer's Responses to Questions

**Comments to the Author**

1. Is the manuscript technically sound, and do the data support the conclusions?

Reviewer #1: Yes

Reviewer #2: Partly

2. Has the statistical analysis been performed appropriately and rigorously? 

Reviewer #1: Yes

Reviewer #2: N/A

3. Have the authors made all data underlying the findings in their manuscript fully available?

Reviewer #1: Yes

Reviewer #2: No

4. Is the manuscript presented in an intelligible fashion and written in standard English?

Reviewer #1: Yes

Reviewer #2: Yes

5. Review Comments to the Author

Reviewer #1: This systematic review aggregated findings from 75 studies that evaluated the validity and reliability of FMS assessment tools for school-aged children. In general, the paper is well-written, informative, and it provides important information on the validity and reliability of FMS assessments. However, the introduction and the methods need clarification because they are too short: for the reader it is hard to follow the sections. The results and the discussion are good.

Abstract:

- explain the abbreviations (TGMD, MABC etc.)

- hard to follow the result section

Introduction:

- too short; needs more details especially in the first section; explain studies in more detail

- the arguementation is not clear

Methods:

- obscure, hard to follow because information on the methods and the results are mixed up

- describe the pre-search step by step

- where are the inclusion and exclusion criteria and their justification?

- why did you not followed a modified PICOS approach?

Results:

- why did you considered children with disorders? needs to be justified in the methods

- table 2 & 3.4 should be a part of the methods

- list the values of the validity and reliability in tables 4-7

Reviewer #2: This article focuses on a very important topic and is of acceptable quality. But to improve the article, there are the following suggestions:

1- What is the main problem of the research statement? Why are researchers focused on observational tools? This is not in the introduction.

2- Search terms are very limited. There are many articles or tools that do not have the term "fundamental or skills" in their titles. This strategy may limit the number of attained articles.

3- Quality evaluation of selective researches is required. This index can show the quality of any research and illustrate the accuracy and dependability of findings. There are many ways and means to evaluate the quality of research.

4- Data extraction criteria have been taken arbitrarily. Using the standard method for evaluation and review can be understandable and useful. The Consensus-Based Standards for the Selection of Health Status Measurement Instruments (COSMIN) is an internationally accepted manner to measurements assessment. Thus this is suggested that the results section of the article arrange according to the COSMIN checklist. This can easily and clearly illustrate the pros and cons of each measurement. This matter also can show the pitfalls and disadvantages of tools collectively. It is not only beneficial in the selection of instruments for clinical practice but also shows issues that are necessary to be considered in new studies for revising and development of new measures.

5- The vast amount of discussion - especially in the beginning - is the paragraph of the result. Results ought to be discussed, not to repeat.

6- Fortunately, in the final sections of the discussion, the authors return to the discussion, and from a broad perspective, they talk about the shortcomings and opportunities of the tools and offer useful and consciousness-raising advice to clinicians and researchers. I believe that this approach should be the dominant frame of discussion and that authors should integrate the findings with theoretical literature, and finally guide clinics, schools, and institutions to gain insight into the problems in this area.

Kind Regards

6. PLOS authors have the option to publish the peer review history of their article (what does this mean?). If published, this will include your full peer review and any attached files.

Reviewer #1: No

Reviewer #2: Yes: Saeed Akbari-Zardkhaneh

---

## [Author Response · Author response to Decision Letter 0]

8 Jul 2020

Dear Dr Montazeri

PONE-D-20-03384: “The validity and reliability of observational assessment tools available to measure fundamental movement skills in school-age children: a systematic review”. 

I would like to begin by thanking yourself and the reviewers for the extremely useful comments on the first submission of our manuscript. We are grateful for being given the opportunity to submit a revised version. 

We have addressed each of the points raised by the reviewers and feel the changes enhance the novel contribution of this review. All edits have been made using tracked changes. 

We have addressed each comment in turn below – the page references refer to the document in ‘simple markup’ format, unless otherwise stated. 

Thank you again for your help with this submission, and we hope that this revised version is suitable for publication in PLOS One.

Yours sincerely,

Lucy Eddy

(On behalf of all co-authors)

Journal Requirements: 

Authors’ comment: The manuscript has been updated to ensure it meets PLOS ONE style requirements. 

• We noticed that the search of your systematic review was last performed in January 2019. Please ensure that the search is up to date and that your systematic review includes any new studies published since then

Authors’ comment: We thank you for the opportunity to include newly published studies. Both the pre-search, and the full search were updated on 19th May 2020. One additional assessment tool, and fifteen additional articles have been added as a result.

Reviewer #1 Comments

1. This systematic review aggregated findings from 75 studies that evaluated the validity and reliability of FMS assessment tools for school-aged children. In general, the paper is well-written, informative, and it provides important information on the validity and reliability of FMS assessments. However, the introduction and the methods need clarification because they are too short: for the reader it is hard to follow the sections. The results and the discussion are good.

 Authors’ comment: We thank the reviewer for their positive comments and have sought to address the concerns about the brevity of some sections (see response to comments 5 through 11).

Abstract:

2. Explain the abbreviations (TGMD, MABC etc.

Authors’ comment: The abbreviations have been described in the abstract (page 2).

3. hard to follow the result section

Authors’ comment: This is a very valuable comment. The results of the abstract have been changed to minimise the number of assessment tools described, in the hope that this will make the results section more clear. Instead, the abstract now describes in more detail the results of the three assessment tools with the most psychometric properties measured. There is also a more general statement about the remaining assessment tools and the feasibility of using current assessments in schools (page 2).

Introduction:

4. too short; needs more details especially in the first section; explain studies in more detail

Authors’ comment: The relationship between fundamental movement skills and other aspects of childhood development have now been clarified in the introduction (page 3, paragraph 1).

5. the arguementation is not clear

Authors’ comment: We agree that the introduction did not make a clear enough argument. The order and flow of the introduction have been edited to explain: i) why FMS are important, ii) issues with identifying children through current healthcare systems, iii) potential benefits of moving assessments into schools and iv) that there are a large number of assessment tools available for use, which makes it difficult for schools to know which is most suitable (pages 3 &4).

Methods:

6. obscure, hard to follow because information on the methods and the results are mixed up

7. describe the pre-search step by step

Authors’ comment: The pre-search now has a full explanation of the search process (page 5, paragraph 1) and the results of the pre-search now feature at the start of the results section (pages 8 & 9, ‘assessment tools’). We would like to thank the reviewer for these observations, which we believe has improved the methods and results sections, making them more clear and distinct. 

8. where are the inclusion and exclusion criteria and their justification?

9. why did you considered children with disorders? Needs to be justified in the methods

Authors’ comment: The inclusion criteria have now been clarified and expanded. This includes a justification of why studies that sampled children with disorders were included (pages 5 - Inclusion Criteria and Preliminary Systematic Search). 

10. why did you not followed a modified PICOS approach?

Authors’ comment: A PICOS approach was not thought to be appropriate for this review because it was not a systematic review of a healthcare interventions. It didn’t look to review experimental Studies of a specific design that evaluated differences between an Intervention or Control and thus only two of the five sections (Population and Outcome) would have been relevant. Instead, we did a pre-search on existing assessment tools, to generate a comprehensive set of search terms defining our Outcome. The search terms we used for age could be thought of as defining our Population. We also utilised a combination of search terms from other validity and reliability systematic reviews to ensure all relevant results were found.

Results:

11. table 2 & 3 should be a part of the methods

Authors’ Comment: Table 2 has been moved to the methods (and is now Table 1 on page 7). Now that the results of the pre-search have been moved to the results section (see response to comment 8), we feel that table 3 should remain in the results section, as it details the number of studies found on the validity and reliability of each assessment tool, as well as the psychometric properties assessed by included studies (pages 12-20). 

12. list the values of the validity and reliability in tables 4-7

Authors’ Comment: We have taken into consideration the reviewer comments about putting statistics into tables 4-7, however, due to the large variation of statistical techniques used, and psychometric properties measured, we feel this information is better placed in the supplementary material 2 to avoid a ‘cluttered’ table that is difficult to decipher. Additionally, for many of the studies in these tables, there wasn’t a singular metric as many included statistics for individual tasks, or subtests. The colours in these tables represent the mean value of all FMS tasks/ subtests in a given paper. This has been clarified in the methods section (page 8 paragraph 1). The values for each classification (poor –excellent) for type of statistical test have also been included as a footnote for each table, for ease of interpretation (tables 4-7). The full breakdown of validity and reliability values is available in supplementary material 2, and this is more clearly explained in the methods section.

Reviewer #2 comments:

13. This article focuses on a very important topic and is of acceptable quality

Authors’ comment: We thank the reviewer for recognising the need for a systematic review on this subject area. 

14. What is the main problem of the research statement? Why are researchers focused on observational tools? This is not in the introduction.

Authors’ comment: We agree that the introduction did not make it clear enough why the focus was on observational assessment tools. This has now been expanded, based on research which suggests that these assessments may be the most feasible for use in schools. This also helps to articulate the main aim of our research, to help discern which of these observational assessments might be viable for use as in-school screening tools, due to having sufficient validity, reliability and feasibility (final paragraph on page 3, continuing onto page 4). 

15. Search terms are very limited. There are many articles or tools that do not have the term "fundamental or skills" in their titles. This strategy may limit the number of attained articles.

Authors’ comment: The search terms for the pre-search were intentionally limited. We only wanted to assess the validity and/or reliability of assessment tools that have been used in peer-reviewed research to assess fundamental movement skills (also known as fundamental motor skills). There are a larger number of assessment tools available to measure motor competence more generally, however, this review focused on FMS specifically. Our aim was to evaluate tools that clearly self-identify specifically as measures of FMS, and thus presumably market themselves to schools as such. 

16. Quality evaluation of selective researches is required. This index can show the quality of any research and illustrate the accuracy and dependability of findings. There are many ways and means to evaluate the quality of research.

Data extraction criteria have been taken arbitrarily. Using the standard method for evaluation and review can be understandable and useful. The Consensus-Based Standards for the Selection of Health Status Measurement Instruments (COSMIN) is an internationally accepted manner to measurements assessment. Thus this is suggested that the results section of the article arrange according to the COSMIN checklist. This can easily and clearly illustrate the pros and cons of each measurement. This matter also can show the pitfalls and disadvantages of tools collectively. It is not only beneficial in the selection of instruments for clinical practice but also shows issues that are necessary to be considered in new studies for revising and development of new measures.

Authors’ comment: We would like to thank the reviewer for suggesting that we use the COSMIN checklists to evaluate study quality, and we think that this addition has significantly improved the manuscript. We have removed the Risk of Bias (ROBANS tool) from the manuscript, and all studies were evaluated using the more fitting COSMIN guidelines instead. The methodology for using this quality assessment can be found within ‘Data Extraction Process & Quality Assessment’ on page 6. A general paragraph on the quality of included studies has been included on page 10 (COSMIN Quality Assessment), and an evaluation of how study quality impacted upon the results of studies can be found within each results section (pages 21-32). The results of the quality assessment for individual studies can also be found in Supporting information 2 – study table (methodological quality column).

17. The vast amount of discussion - especially in the beginning - is the paragraph of the result. Results ought to be discussed, not to repeat.

Authors’ comment: We agree that the start of the discussion was repetitive of the results (see the final paragraph on page 33 of the ‘all markup’ version of the manuscript). We have edited the discussion, so the introductory paragraph is a brief overview of results, before discussing the results in conjunction with relevant literature.

18. Fortunately, in the final sections of the discussion, the authors return to the discussion, and from a broad perspective, they talk about the shortcomings and opportunities of the tools and offer useful and consciousness-raising advice to clinicians and researchers. I believe that this approach should be the dominant frame of discussion and that authors should integrate the findings with theoretical literature, and finally guide clinics, schools, and institutions to gain insight into the problems in this area.

Authors’ comment: This was a very valuable comment. The discussion has now been modified so that this is the main focus. We have restructured the discussion to make the argument clearer. The discussion now explains i) from the systematic review, which assessment tool was the most valid and reliable, and how suitable this assessment tool is for use within schools, based on feasibility criteria, ii) the type of assessment tool most feasible for use in schools, and justification as to why, iii) the psychometric properties of assessment tools which fell within this group (pages 32-35).

---

## [Editor Report · Decision Letter 1]

6 Aug 2020

The validity and reliability of observational assessment tools available to measure fundamental movement skills in school-age children: a systematic review.

PONE-D-20-03384R1

Dear Dr. Eddy,

We’re pleased to inform you that your manuscript has been judged scientifically suitable for publication and will be formally accepted for publication once it meets all outstanding technical requirements.

Kind regards,

Ali Montazeri

Academic Editor

PLOS ONE
---

## [Editor Report · Acceptance letter]

12 Aug 2020

PONE-D-20-03384R1 

The validity and reliability of observational assessment tools available to measure fundamental movement skills in school-age children: a systematic review. 

Dear Dr. Eddy:

I'm pleased to inform you that your manuscript has been deemed suitable for publication in PLOS ONE. Congratulations! Your manuscript is now with our production department. 

Kind regards, 

on behalf of

Professor Ali Montazeri 

Academic Editor

PLOS ONE